# Mediating effects of general self-efficacy on the relationship between the source of meaning in life and prosocial behaviours in vocational college nursing students: A cross-sectional study

Linan Cheng[1☯], Yansheng Ye[1☯¤], Zhaoyang Zhong[2☯], Fengying Zhang[1‡], Xiuying Hu[1‡], Renshan Cui[3☯]*, Qian Chen[1☯]*

1 West China School of Nursing, West China Hospital, Sichuan University, Chengdu, Sichuan, China,
2 Fenyang College of Shanxi Medical University, Fenyang, Shanxi, China, 3 Jiaxing University College of Medicine, Jiaxing, Zhejiang, China

☯ These authors contributed equally to this work.
¤ Current address: The People's Hospital of Yuxi City, The Sixth Affiliated Hospital of Kunming Medical University, Yuxi, Yunnan, China
‡ These authors also contributed equally to this work.
* 2594793790@qq.com (RC); chen_qian@scu.edu.cn (QC)

## Abstract

### Objectives

To examine the mediating effect of general self-efficacy on the relationship between the source of meaning in life (SML) and prosocial behaviours in vocational college nursing students.

### Methods

Between March and June 2019, a cross-sectional descriptive study was conducted, and 799 nursing students from three vocational colleges completed the Source of Meaning in Life Scale, General Self-Efficacy Scale and Prosocial Behaviours Scale. Data were analyzed using structural equation modelling and statistical analysis by SPSS (version 23.0, IBM).

### Results

The average SML, general self-efficacy and prosocial behaviours scores of the 799 nursing students were 6.43±0.83, 2.48±0.59 and 3.69±0.62, respectively. Correlation analysis showed that SML, general self-efficacy and prosocial behaviours were positively correlated (P<0.01). General self-efficacy partially mediated the relationship between SML and prosocial behaviours (P<0.01); this mediating effect contributed 22.97% of the total effect and explained 17.6% of the variance in the dependent variable.

**Data Availability Statement:** All relevant data are within the paper and its Supporting information files.

**Funding:** This work was supported by Science & Technology Department of Sichuan Province,in the framework of assessment of symptoms and establishment of a multidisciplinary palliative care model for elderly patients with terminal frailty (2019YFS0386). The person in charge of the project is corresponding author Qian Chen in the manuscript. And there was no additional external funding received for this study.

**Competing interests:** The authors have declared that no competing interests exist.

## Conclusions

Educators should focus on cultivating nursing students' cognition and experience of meaning in life and their efficacy in life, study and work, which can improve students' "people-oriented" service and prosocial behaviour and the quality of nursing services.

## Introduction

In China, 3-year vocational college nursing students constitute an important part of the nursing work force and account for more than 60% of the total number of nursing students [1, 2]. "The national health and family planning talent development plan for the 13th five-year plan period in China" clearly called for the expansion of the scale of vocational training as a starting point for nursing personnel and the improvement of the humanistic care quality among nursing students to further strengthen professional and technical talents in health and family planning and to meet the increasing medical consumption [3]. Compared with the curricula of nursing colleges in other countries, the humanistic nursing curriculum in China was developed late, and it has not been universally applied [4]. In some colleges, the humanistic nursing curriculum has a low utilization rate and less content than in other colleges, and the forms and methods of teaching need to be improved. As part of the future nursing force, nursing students should have a certain level of humanistic care quality, an ability to adapt to changes in medical models and people's needs and an ability to promote the development of their nursing careers.

As reported, vocational college students are generally between the ages of 17 and 24 and are mostly new graduates from middle or high school. The majority of them are from only-child families, female, and unmarried [1]. They have relatively unique advantages but also lack positive motivation, have poor communication skills, lack a strong sense of self, have weak teamwork ability, and have poor self-bearing and poor self-management ability [5, 6]. In addition, these students are in an important period in the formation of their life and professional values, so it is very important to cultivate their knowledge and skills, provide them with a quality education and cultivate their humanistic care ability [7].

A source of meaning in life (SML) refers to an important, valuable event that people have experienced in their lives, and it is an important factor for individuals to perceive the value and meaning of life [8, 9]. The perception of SML can make the individual feel that life is full of meaning. In contrast, an individual's failure to find meaning or purpose in existence can lead to feelings of emptiness, neurosis, substance abuse, and even suicide, which are negative effects that accompany a lack of "meaning" [10–13]. Prosocial behaviour refers to altruistic behaviours that are consistent with social expectations and bring benefits to others, the collective, society and the country [14]. Studies show that adolescents with higher prosocial behaviour are more responsible and sympathetic than those without such behaviour, are good at perspective taking, and have relatively high levels of prosocial moral reasoning [14–16]. Some studies show that college students' sense of meaning in life has a significant predictive effect on prosocial behaviors [17, 18]. As an important internal psychological resource, general self-efficacy is necessary for individuals to perceive meaning in life, and it can enhance individuals' positive life experience [9, 15] and prosocial behaviour [19–23]. Some researches also have shown that greater self-efficacy can predict lower impulsivity and better prosocial behavior for youth [24, 25]. Self-efficacy can explain the level of prosocial organizational behaviour [26]. And some literature has explored the mediating effect of life meaning of college students(or adolescents) between family cohesion [27], nostalgia [28], the moral sense of life [17] and post-

traumatic stress disorder [18] and prosocial behaviors. While few researches explore the mediating effect of self-efficacy, the source of meaning in life (SML) and prosocial behaviors. However, there is indeed some correlation between them according to the contents of the literature.

Vocational college nursing students represent the main component of the nursing workforce, and thus, their physical and mental development, career development and career recognition in nursing directly affect the balance and development of the nursing profession.

According to the knowledge, attitude, belief, practice model [29–31] and motivation theory [32], SML can be regarded as a form of endogenous motivation and enhance the faith and values of vocational college nursing students, thus improving their altruistic prosocial behaviour.

Based on previous literature, we developed a hypothetical model and predicted that general self-efficacy would have a mediating effect between SML and prosocial behaviours (Fig 1). Therefore, this study aimed to examine the mediating effect of general self-efficacy on the relationship between SML and prosocial behaviours in vocational college nursing students.

## Methods

### Study design and ethical considerations

A cross-sectional design was used in this study. The STROBE cross-sectional reporting guidelines were used. Ethical approval of this study was granted by the West China Hospital of Sichuan University Biomedical Research Ethics Committee (ethics number: 2016–272). Ethical approval included the approval of the consent procedure and the written informed consent form. Participants voluntarily and anonymously participated in this study. All data collected were kept confidential and used only for this research study.

### Participants

From March to June 2019, a total of 799 respondents from three colleges in three provinces (Shanxi, Zhejiang and Sichuan) were selected by convenience sampling. Students who met the following inclusion criteria were included: vocational college nursing students who a)

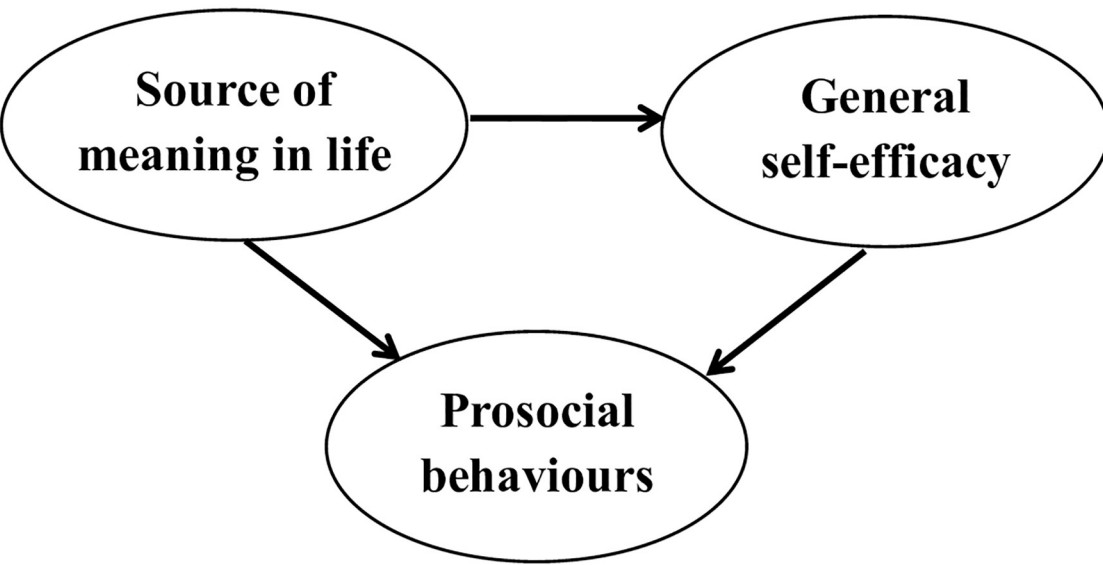

**Fig 1. Hypothesized theoretical model.**

provided informed consent to participate and b) had not yet graduated. The exclusion criteria were as follows: nursing students who a) were on sick leave during the investigation or b) had transferred to another specialty or had been transferred from another specialty to nursing less than six months prior to the study.

## Instruments

The questionnaires included instruments assessing participants' demographic characteristics (age, gender, grade, residence, religion, and only-child status), general self-efficacy, source of meaning in life and prosocial behaviours.

The General Self-Efficacy Scale (GSES) was developed by Schwarzer and colleagues [33]. Based on the original version, Wang and colleagues created a new GSES suitable for Chinese people that includes 10 items [34]. Responses are provided on a 4-point scale from 1 (incorrect) to 4 (completely correct), with the total possible score ranging from 10 to 40 points. A higher score indicates stronger general self-efficacy. This scale is a reliable tool for measuring nurses' general self-efficacy. In this study, the Cronbach's alpha coefficient was 0.903.

The Source of Meaning in Life Scale (SMLS) was developed by Cheng MM and colleagues [8, 9]. This scale consists of 30 items categorized into five subscales: social concern (9 items), self-growth (8 items), relationship harmony (6 items), enjoyment in life (4 items) and physical and mental health (3 items). Responses are provided on a seven-point scale from 1 (not at all meaningful) to 7 (totally meaningful). The Cronbach's coefficient of the total scale was 0.94. In this study, the Cronbach's alpha coefficient was 0.978.

The Prosocial Behaviours Scale (PBS) was compiled by Carlo et al. in 2002 [15]. Kou and colleagues revised the scale to be suitable for Chinese people [35]. This revised scale consists of 26 items categorized into six subscales: open (4 items), anonymous (5 items), altruistic (4 items), compliant (5 items), emotional (5 items) and urgent (3 items). Responses are provided on a scale ranging from 1 (very unlike me) to 7 (very like me). The Cronbach's coefficients of each dimension of the scale ranged from 0.494–0.778. In this study, the Cronbach's alpha coefficient was 0.962.

## Quality control

Before the survey, we selected a college class for a preliminary investigation and modified the questionnaire. Meanwhile, we conducted unified training to avoid bias caused by the investigators. To obtain participants' informed consent, we informed all participants that their responses were anonymous and that they would experience no consequences as a result of their participation in or withdrawal from the study. We took adequate measures to minimize missing data. The ad hoc imputation method was used when there was a small amount of missing data (10%). In this study, EpiData 3.1 (The EpiData Association, Denmark) was used to input data to ensure the accuracy of the data.

## Data collection

The data were collected between March and June 2019. We selected three colleges in Shanxi, Zhejiang and Sichuan Provinces in China by using convenience sampling. After the approval of college managers and instructors was obtained, three trained research assistants visited the colleges and briefly explained the purpose of the study. Participants involved in the study provided informed consent and spent 15~20 minutes completing the questionnaires. Among the 820 returned questionnaires, 21 were incomplete due to missing values or consistently repeated response options. Therefore, 799 questionnaires were suitable for analysis, for an effective response rate of 97.43%. More than 200 samples must be obtained for structural

equation model analysis [36]. Therefore, the inclusion of 799 vocational college respondents was valid.

## Data analyses

**Structural equation model analysis.** We used analysis of moment structures to estimate the hypothetical model and the maximum likelihood (ML) method to confirm the relationships and predictions (AMOS, version 23.0). Bootstrapping was used to address the non-normal distribution of samples using the ML method. The measurement errors of the predictions were also considered. The goodness of fit standard for the structural equation model was mainly assessed by absolute fit indices (RMSEA<0.08, GFI>0.09 and AGFI>0.09), relative fit indices (NFI>0.90, CFI>0.90 and RFI>0.90), and simple fit indices (PGFI>0.50, PNFI>0.50, PCFI>0.50 and $X^2/df$ <5.00). It is necessary to modify the model to improve fit when modification indices are larger than ten [37, 38].

**Primary analysis.** The general self-efficacy, SML and prosocial behaviours scores were used to calculate descriptive statistics and conduct t-tests, one-way analysis of variance (ANOVA), and multiple linear regression by SPSS 23.0 (IBM Corp, Armonk, NY, USA). Statistical significance was set at P < .05, two-tailed.

## Results

### Demographic characteristics

A total of 799 nursing students participated in the survey. Participants had a mean age of 19.71 years (SD = 1.63; range = 17–25). Among the respondents, 91.1% were female (n = 728), 75.30% were ethnically Han (n = 602) and 53.30% were junior students (426). Additionally, 46.5% came from urban cities, 56.3% were from rural cities and 81.10% had no religion. Only 11.90% reported a religion, including Christianity, Catholicism and Buddhism. Of the participants, 27.20% were only children, while 72.80% had siblings. Ethnic minorities reported better prosocial behaviours than Han participants (t = 9.993, p = 0.002), and participants who lived in urban areas reported better prosocial behaviours than those who lived in rural areas (t = 9.018, p = 0.003) (Table 1).

**General self-efficacy, SML and prosocial behaviours.** The mean SML score was above the moderate level (M = 6.43, SD = 0.83). Among the SML subscale scores, the highest score was for physical and mental health, while the lowest score was for social concern. The mean prosocial behaviour score was at an intermediate level (M = 3.69, SD = 0.62). The ranking of the subscale scores from high to low was as follows: urgent, altruistic, compliant, emotional, anonymous and open. The mean self-efficacy score was relatively low (M = 2.48, SD = 0.59) (Table 2).

Prosocial behaviours had significant positive correlations with general self-efficacy (r = 0.488, P < 0.01) and SML (r = 0.354, P < 0.01), which means that as self-efficacy and SML scores increase, prosocial behaviours scores also increase (Table 3).

### Moderating effect of general self-efficacy on SML and prosocial behaviours

The hypothetical model showed no negative variance and a large standard error in the model, meaning that the model did not violate the identification rule [36] and the hypothesis model was modified according to the calculation results. The absolute fit indices of the model ($X^2/df$ = 3.546, GFI = 0.968, AGFI = 0.944, NFI = 0.984, RFI = 0.976, CFI = 0.988, PNFI = 0.671, PGFI = 0.5581, PCFI = 0.674, RMSEA = 0.056) all indicate the adequate fit to the data (Fig 2). SML was positively related to general self-efficacy (B = 0.21, P < 0.01), general self-efficacy was

**Table 1. Demographic characteristics and differences in prosocial behaviours (N = 799).**

| Variable | | N(%) | Mean±SD | t(F) | P |
|---|---|---|---|---|---|
| Gender | Male | 71(8.9) | 96.11±15.62 | 0.203 | 0.880 |
| | Female | 728(91.1) | 95.81±16.240 | | |
| Ethnicity | Han | 602(75.3) | 94.81±15.07 | 9.993 | 0.002 |
| | Ethnic minority | 197(24.7) | 98.98±18.87 | | |
| Grade | Freshman year | 124(15.5) | 95.55±15.31 | 0.257 | 0.773 |
| | Sophomore year | 249(31.2) | 95.33±15.38 | | |
| | Junior year | 426(53.3) | 96.21±16.88 | | |
| Residence | Urban | 349(43.7) | 97.78±16.83 | 9.018 | 0.003 |
| | Rural | 450(56.3) | 94.33±15.50 | | |
| Religion | Yes | 95(11.9) | 94.49±16.13 | 0.740 | 0.390 |
| | No | 704(88.1) | 96.02±16.19 | | |
| Family with only child | Yes | 217(27.2) | 96.05±16.02 | 0.051 | 0.822 |
| | No | 582(72.8) | 95.76±16.25 | | |
| Age(year) | ≦21 | 675(84.5) | 95.93±16.27 | 0.142 | 0.706 |
| | ≧22 | 124(15.5) | 95.33±16.18 | | |

Note. Abbreviation: N = number SD = standard deviation

$P < 0.05$.

positively related to prosocial behaviours (B = 0.43, P < 0.01), and SML was related to prosocial behaviours (B = 0.26, P < 0.01). These findings suggest that general self-efficacy had a partial mediating effect between SML and prosocial behaviours among vocational college nursing students; this mediating effect contributed 22.97% to the total effect and explained 17.6% of the variance in the dependent variable (Table 4).

**Table 2. The scores among general self-efficacy, the source of meaning in life and prosocial behaviours (Mean ± SD, N = 799).**

| Items | The Number of Items | The Score of Items | The Mean Score of Items |
|---|---|---|---|
| SML | 30 | 192.90±24.91 | 6.43±0.83 |
| PMH | 3 | 19.56±2.66 | 6.52±0.89 |
| RH | 6 | 38.98±4.96 | 6.50±0.83 |
| SG | 8 | 51.92±7.00 | 6.49±0.88 |
| EL | 4 | 25.61±3.65 | 6.40±0.91 |
| SC | 9 | 56.83±8.14 | 6.31±0.90 |
| PB | 26 | 95.83±16.18 | 3.69±0.62 |
| Urgent | 3 | 11.45±2.18 | 3.82±0.73 |
| Altruistic | 4 | 15.19±2.96 | 3.80±0.74 |
| Compliant | 5 | 18.71±3.40 | 3.74±0.68 |
| Emotional | 5 | 18.45±3.45 | 3.69±0.70 |
| Anonymous | 5 | 18.06±3.56 | 3.61±0.71 |
| Open | 4 | 13.97±2.94 | 3.49±0.74 |
| SE | 10 | 24.83±5.90 | 2.48±0.59 |

Note. Abbreviations: The source of meaning in life = SML;PMH = physical and mental health; RH = relationship harmony;EL = Enjoy in Life; SC = social concern;SG = self-growth; PB = prosocial behaviours;SE = Self-efficacy.

**P < 0.01,

*P < 0.05.

**Table 3.  Correlations among general self-efficacy, the source of meaning in life and prosocial behaviours (N = 799).**

| Variable | SML | PMH | RH | EL | SC | SG | PB | Open | Anonymous | Altruistic | Compliant | Emotional | Urgent | SE |
|---|---|---|---|---|---|---|---|---|---|---|---|---|---|---|
| SML | 1 | | | | | | | | | | | | | |
| PMH | .884** | 1 | | | | | | | | | | | | |
| RH | .945** | .840** | 1 | | | | | | | | | | | |
| EL | .936** | .802** | .857** | 1 | | | | | | | | | | |
| SC | .944** | .766** | .854** | .844** | 1 | | | | | | | | | |
| SG | .967** | .862** | .896** | .915** | .861** | 1 | | | | | | | | |
| PB | 0.354 | .292** | .318** | .310** | .383** | .317** | 1 | | | | | | | |
| Open | .233** | .139** | .205** | .218** | .287** | .186** | .772* | 1 | | | | | | |
| Anonymous | .293** | .252** | .266** | .249** | .314** | .262** | .870** | .546** | 1 | | | | | |
| Altruistic | .336** | .312** | .303** | .288** | .346** | .309** | .893* | .543* | .546** | 1 | | | | |
| Compliant | .314** | .261** | .278** | .269** | .339** | .287** | .910** | .645** | .736** | .797** | 1 | | | |
| Emotional | .351** | .293** | .319** | .314** | .374** | .312** | .917** | .702** | .742** | .776** | .789** | 1 | | |
| Urgent | .334** | .273** | .300** | .294** | .351** | .311** | .880** | .635** | .698** | .788** | .790** | .780** | 1 | |
| SE | .222** | .163** | .189** | .191** | .246** | .208** | .488** | .389** | .443** | .410** | .407** | .479** | .422** | 1 |

Note. Abbreviations: The source of meaning in life = SML;PMH = physical and mental health; RH = relationship harmony;EL = Enjoy in Life; SC = social concern;

SG = self-growth; PB = prosocial behaviours;SE = Self-efficacy.

**P < 0.01,

*P < 0.05.

## Discussion

Our main aim was to verify the predictive ability of SML and self-efficacy for prosocial behaviours. We found that SML and general self-efficacy were significant predictors, but they worked through different mechanisms: general efficacy had a direct effect, while SML had an indirect effect that was partially mediated by general efficacy. Thus, we have a deeper understanding of the value and function of SML for vocational college nursing students. Additionally, our understanding of the mechanisms of self-efficacy in practical applications is extended. It could be argued that these findings indicate that there is a better way to improve prosocial behaviours in students and the quality of nursing service and to meet the health needs of the public.

In the study, the mean SML score was above the median level, which implied that physical and mental health, relationship harmony, enjoyment in life, social concern and self-growth were ways to find meaning in life. The highest subscale score was for physical and mental health, which is in line with the attention to the physical and mental health of students among society and students' families. In addition, the mean score for relationship harmony was consistent with the finding of Sheng et al. (2007) [39]. This result is consistent not only with previous findings on the trajectory of the growth and development of interpersonal relationships among Chinese nursing students in vocational colleges but also Maslow's work on needs for love and a sense of belonging. Therefore, we should actively develop the humanistic nursing curriculum, strengthen nursing students' interpersonal communication skills and develop their social functions. Nursing students are in a stage of learning and development and may lack the motivation and resources to grow, so they might not have fully recognized and achieved their self-growth; therefore, it is necessary to strengthen the career planning education of nursing students. The improvement of living standards and learning environments would facilitate their pursuit of enjoyment in life. Social concern refers to individuals' willingness to contribute to society and to pay attention to and respect others; in this study, we hoped

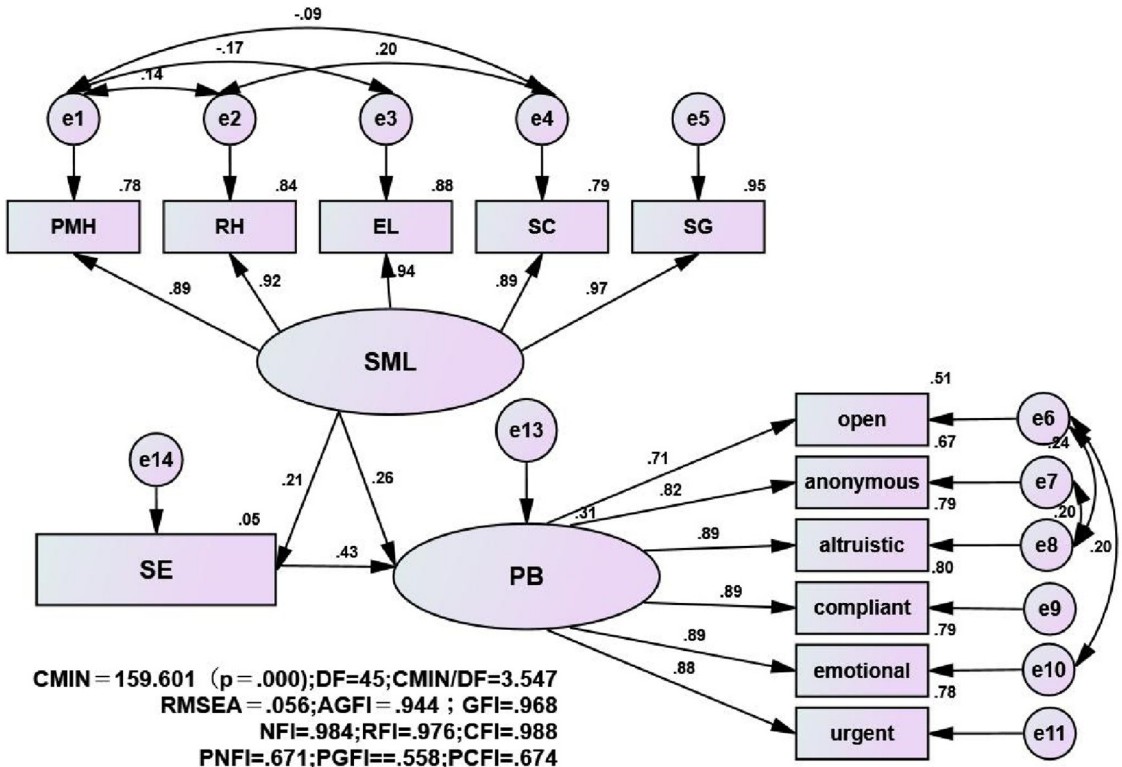

**Fig 2. Results of the effect of self-efficacy on the source of meaning in life and prosocial behaviours among the whole sample.**
Note. Abbreviations: PMH = physical and mental health; RH = relationship harmony; EL = Enjoy in Life; SC = social concern; SG = self-growth; PB = prosocial behaviours; The source of meaning in life = SML; SE = Self-efficacy. All the coefficients in this figure are standardized and significant at level 0.01. The numbers on the straight arrows indicate the standard path coefficients.

to find evidence of social concern among the nursing students, but the social concern subscale had the lowest mean score. Therefore, we should pay attention to the improvement and development of education in humanistic care, values and morality. In this study, the students' self-efficacy scores indicated that we should also pay attention to the improvement of students' self-confidence.

**Table 4. Regression of mediating effect among general self-efficacy, the source of meaning in life and prosocial behaviours (N = 799).**

| Variable | Model 1 Prosocial behaviours (dependent variable) | | | Model 2 General self-efficacy (dependent variable) | | | Model 3 Prosocial behaviours (dependent variable) | | |
|---|---|---|---|---|---|---|---|---|---|
| | B | SE | β(CI) | B | SE | β(CI) | B | SE | β(CI) |
| The source of meaning in life | 0.230 | 0.022 | 0.354 (0.188~0.272) | 0.053 | 0.008 | 0.222 (0.037~0.069) | 0.168 | 0.020 | 0.259 (0.129~0.207) |
| General self-efficacy | | | | | | | 1.179 | 0.083 | 0.430 (1.016~1.343) |
| R | | 0.354 | | | 0.222 | | | 0.549 | |
| Adjusted R² | | 0.124 | | | 0.048 | | | 0.300 | |
| F | | 114.212 | | | 41.282 | | | 171.642 | |
| P | | 0.000 | | | 0.000 | | | 0.000 | |

Note: β coefficients in this figure are standardized. Effect M = 0.430*0.222/0.354 = 0.2297; The mediating effect explained that the variance variation of the dependent variable was sqrt (0.300–0.124) = 0.176(17.6%).
**P<0.01.

The total mean prosocial behaviour score of vocational college nursing students and all subscale scores reached a relatively ideal level; the highest mean subscale score was for the urgent subscale, while the lowest score was for the public subscale. On the one hand, these results indicated that the prosocial behaviours of vocational college nursing students were more susceptible to urgent situations or emotional factors [40]. On the other hand, they showed that the nursing students all had good altruistic behaviours. In the face of a crisis, they would all be able to show their noble professional dedication to respecting life. The open subscale refers to prosocial tendencies implemented in the presence of others; the scores for this subscale further suggested that the nursing students were not willing to disclose their helpful behaviour, which reflects the fact that people who engage in social avoidance are more reluctant to engage in certain behaviours in public [41].

The finding that ethnic minorities had better prosocial behaviours than Han participants was related to the cultural environment, suggesting that students from ethnic minorities will communicate with mainland students more actively to better adapt to the learning and living environment and will in turn have better interpersonal skills and prosocial behaviours. Participants who lived in urban areas reported better prosocial behaviours than those who lived in rural areas, which may be related to their educational backgrounds. Additionally, rural students may lack interpersonal security, and their prosocial behaviours may be influenced by their rural identity.

Our results suggest that general self-efficacy partially mediated the relationship between SML and prosocial behaviour. In other words, the cognition of SML among vocational nursing students not only directly predicts prosocial behaviours but also indirectly predicts general self-efficacy. If individuals lack meaning in life, they will have a more antisocial psychology and behaviour problems according to logotherapy theory [42, 43]. However, if individuals have accurate cognition and comprehension of meaning in life, they will actively devote themselves to contributing to society, better understand the existence of others and the significance of other people's lives, and thus deliver humanistic nursing services that meet the needs of society [44], which may promote the development of good doctor-patient relationships. Therefore, as a dynamic motivating factor, SML can directly promote prosocial behaviours [45]. Pearson correlation analysis showed that there was a positive correlation between SML, general self-efficacy and prosocial behaviours in the study. Students with SML may have a higher sense of self-efficacy than those without SML, and a higher sense of self-efficacy is conducive to prosocial behaviours. Although different scholars and ideologists have different views on SML, they all believe that SML is a subjective consciousness that is influenced by one's cultural background and is related to real life [46]. Studies show that SML endows individuals with meaning in life. According to a relativist perspective, if individuals believe that life is meaningful, they will form the belief that "life is meaningful" and will form their own goals and values in life [9], so these individuals can better adapt to society and engage in more prosocial behaviours than those who do not believe that life is meaningful. The level of general self-efficacy determines the individual's attitude and actions related to prosocial behaviours [47]. Individuals with a high sense of self-efficacy have not only clear goals and directions for their own lives but also a sense of individual value and control; they will actively interact with society.

The latest view proposes a multidimensional concept of a continuum of cognitive, emotional and behavioural tendencies [48]. Nursing students' exploration and cognition of SML allow them to find purpose and meaning in life and experience emotions related to meaning in life; these emotions refer to one's satisfaction and sense of achievement obtained from past experience and a completed goal [49]. Such emotions not only strengthen the individual's confidence in the face of difficulties but also have an internal effect on the individual's behaviour.

Related research results also show that people with a higher sense of self-efficacy tend to perform more altruistic behaviours, such as sharing, offering and helping others [50].

Vocational college nursing students are the main part of the workforce who can meet the needs of medical and health services. Therefore, educators should improve training mechanisms and teaching methods through teaching reform or practice; pay attention to the cultivation of nursing students' cognition and experience of meaning in life; develop nursing students' inner strengths in life, study and work; and further enhance students' abilities to provide humanistic nursing service and improve the quality of nursing care. For example, we could improve our teaching methods, such as increasing scene teaching method can improve students' situational coping ability, self-confidence and empathy ability. Meanwhile, we encourage to increase more humanities courses to increase nursing students' awareness of humanistic care and improve the quality of humanistic care. Moreover, some relevant lectures should be hold regularly to enlighten nursing students' critical thinking and improve their confidence through the power of a good example.

This study has several limitations. First, our use of convenience sampling (among vocational college nursing students from three colleges) might limit the generalizability and robustness of the study's results. Therefore, nursing students with diverse educational backgrounds should be included in future research to verify our hypothesis. Second, the self-report questionnaires may not truly reflect the thoughts of nursing students due to the flaws in this method of data collection, but the limitation may not have negatively influenced the results, as we implemented strict quality control. Finally, our conclusions are based on cross-sectional data, meaning we make conclusions about cause-effect relationships between the studied variables. Therefore, a subsequent large-scale, longitudinal investigation is necessary. Nevertheless, we believe that these limitations do not nullify our conclusions.

## Conclusions

We have confirmed the partial mediating effect of general self-efficacy on prosocial behaviours in vocational college nursing students. This finding implies that nursing educators should consider the importance of SML and general self-efficacy for improving the quality of nursing care. Additionally, effective measures should be taken to improve the training mechanisms and teaching methods through teaching reform or practice. While this study may be applicable only to undergraduate vocational college nursing students, our model can be used to improve prosocial behaviours models for nursing students and can provide a foundation for the improvement of theories, such as those on humanistic care or SML, and interventions.

## Supporting information

**S1 Statistics.**
(SAV)

## Acknowledgments

The authors are thankful for the supervisors and 799 vocational nursing students who took part in the study.

## Author Contributions

**Data curation:** Qian Chen.

**Funding acquisition:** Qian Chen.

**Investigation:** Linan Cheng, Yansheng Ye, Zhaoyang Zhong, Fengying Zhang, Xiuying Hu.

**Methodology:** Linan Cheng, Yansheng Ye.

**Resources:** Zhaoyang Zhong, Fengying Zhang, Qian Chen.

**Supervision:** Fengying Zhang, Renshan Cui, Qian Chen.

**Writing – original draft:** Linan Cheng, Zhaoyang Zhong.

**Writing – review & editing:** Renshan Cui, Qian Chen.

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
