## [Decision Letter · Decision Letter 0]

8 Oct 2020

PONE-D-20-15550

Mediating effects of general self-efficacy on the relationship between the source of meaning in life and prosocial behaviours in vocational college nursing students:a cross-sectional study

PLOS ONE

Dear Dr. Chen,

Thank you for submitting your manuscript to PLOS ONE. After careful consideration, we feel that it has merit but does not fully meet PLOS ONE’s publication criteria as it currently stands. Therefore, we invite you to submit a revised version of the manuscript that addresses the points raised during the review process.

Please pay particular attention to Reviewer 1's comments.

We look forward to receiving your revised manuscript.

Kind regards,

Nancy Beam, PhD

Staff Editor

PLOS ONE

Journal Requirements:

This work was supported by Science & Technology Department of Sichuan Province,in the framework of assessment of symptoms and establishment of a multidisciplinary palliative care model for elderly patients with terminal frailty(2019YFS0386).The person in charge of the project is corresponding author Qian Chen in the manuscript.

i) Please provide an amended statement that declares *all* the funding or sources of support (whether external or internal to your organization) received during this study, as detailed online in our guide for authors at http://journals.plos.org/plosone/s/submit-now.  Please also include the statement “There was no additional external funding received for this study.” in your updated Funding Statement.

ii) Please include your amended Funding Statement within your cover letter. We will change the online submission form on your behalf.

Reviewers' comments:

Reviewer's Responses to Questions

**Comments to the Author**

1. Is the manuscript technically sound, and do the data support the conclusions?

Reviewer #1: Partly

Reviewer #2: Yes

2. Has the statistical analysis been performed appropriately and rigorously? 

Reviewer #1: Yes

Reviewer #2: Yes

3. Have the authors made all data underlying the findings in their manuscript fully available?

Reviewer #1: Yes

Reviewer #2: Yes

4. Is the manuscript presented in an intelligible fashion and written in standard English?

Reviewer #1: Yes

Reviewer #2: Yes

5. Review Comments to the Author

Reviewer #1: Thank you for the possibility to review this article. This study is very interesting for nursing educators and the quality of care for patients. However, I have some comments:

1) It could be interesting to have more critical background about general self-efficacy, the source of meaning in life (SML) and prosocial behaviours. Are there previous studies examining these dimensions? In a population of students? Or in Nursing Students? Which were the finding? There are similar models in other studies?

2) Why there is the need to improve the humanistic care quality among nursing students? can you give some information about this sentence? It is not clear in which way the general self-efficacy, the source of meaning in life (SML) and prosocial behaviours are linked with improvement of humanistic care quality among students.

3) Why in introduction about students’ characteristics? I have not clear the reasons for these information.

4) You had hypothesized a relation between general self-efficacy, the source of meaning in life (SML) and prosocial behaviours and for these reasons you had tested the model. But on the basis of what? Are there previous studies?

5) You said “Studies show that adolescents with higher prosocial behaviour are more responsible and sympathetic than those without such behaviour, are good at perspective taking, and have relatively high levels of prosocial moral reasoning” but in your study you are speaking about nursing students and not adolescents. Are there other studies about student’s population? about health care workers?

6) In pag 1, line 31 there are two and.

7) About the instruments there are only few information. GSES come from WANG but in which kind of population was it validated? In students? Nursing students? If not, there is the necessity to validate the GSES in nursing students. Can we have some information about explorative factor analysis with the identification of the dimensions and about confirmative factor analysis? You used the total score of the GSES but there is a secondary order factor? At the same way it should be reported information about SMLS and PBS.

8) In table 1 there are the differences in prosocial behaviour and demographic characteristics. Why we have not the same information about general self-efficacy and SLM?

9) The paper need more critical discussion. Which is the implication of these finding? Can you work on general self-efficacy to improve the humanistic care quality among nursing students? in which way?

10) In pag 15, line 270 there is ERROR??

Thank you very much. I think this study is very interesting and can be improved.

Reviewer #2: Dear Author,

thank you for the opportunity to read such an interesting paper. I agree that it`s very important to strenghten nursing students` interpersonal skills, social functions and career planing education also we shoud pay attantion to the improvement of students` self-confidence.

please check the 270 line what the error means?

and the reference list - the correctness of writing and editing

Good luck

6. PLOS authors have the option to publish the peer review history of their article (what does this mean?). If published, this will include your full peer review and any attached files.

Reviewer #1: No

Reviewer #2: No

---

## [Author Response · Author response to Decision Letter 0]

1 Nov 2020

Dear Editors:

Thank you for your comments concerning our manuscript entitled “Mediating effects of general self-efficacy on the relationship between the source of meaning in life and prosocial behaviours in vocational college nursing students:a cross-sectional study" (PONE-D-20-15550) .We have made correction carefully according to comments.The main corrections in the paper and the responds are as following :

Response to Staff Editor Dr.Nancy Beam 'Comments

1.Please ensure that your manuscript meets PLOS ONE's style requirements, including those for file naming. 

Response: Yes, our manuscript meets PLOS ONE's style requirements.

2.Could you therefore please include the title page into the beginning of your manuscript file itself, listing all authors and affiliations.

Response: Yes,we have included the title page into the beginning of our manuscript file itself, listing all authors and affiliations.

3.We note that you have indicated that data from this study are available upon request. PLOS only allows data to be available upon request if there are legal or ethical restrictions on sharing data publicly. For information on unacceptable data access restrictions, please see http://journals.plos.org/plosone/s/data-availability#loc-unacceptable-data-access-restrictions.

Response: Yes,we will upload the minimal anonymized data set necessary as either Supporting Information files .

4.Funding Statement

Response: Yes,we have provided an amended statement in cover letter and updated Funding Statement.

Reviewers' comments

Reviewer #1:

1. It could be interesting to have more critical background about general self-efficacy, the source of meaning in life (SML) and prosocial behaviours. Are there previous studies examining these dimensions? In a population of students? Or in Nursing Students? Which were the finding? There are similar models in other studies?

Response: Thank you for your recognition and support of my article. As you said, I really should add more critical background .And I have added more critical background about general self-efficacy, the source of meaning in life (SML) and prosocial behaviours in (line 74-82 page 4). Some researches also have shown that greater self‐efficacy can predict lower impulsivity and better prosocial behavior for youth.Self-efficacy can explain the level of prosocial organizational behaviour.And some literature has explored the mediating effect of life meaning of college students(or adolescents) between family cohesion, nostalgia, the moral sense of life and post-traumatic stress disorder and prosocial behaviors. While few researches explore the mediating effect of self-efficacy, the source of meaning in life (SML) and prosocial behaviors. However, there is indeed some correlation between them according to the contents of the literature.

2.Why there is the need to improve the humanistic care quality among nursing students? can you give some information about this sentence? It is not clear in which way the general self-efficacy, the source of meaning in life (SML) and prosocial behaviours are linked with improvement of humanistic care quality among students.

Response: Thank you for your good advice.First of all, strengthening the learning of humanistic knowledge and improving the humanistic quality of nursing students are based on the requirements of national policies and guidelines, such as:The National Health Commission's "Notice on Further Deepening quality Care and Improving Nursing Services" (No. 15 [2015] of the State Health Office) requires us to further strengthen the consciousness of humanistic care and improve the level of nursing services.Therefore, how to deepen the reform of nursing education and strengthen the humanistic quality of nursing has become one of the focuses of nursing educators and managers.

Second, the development of prosocial behavior of nursing students conforms to the concept and essence of human concern in nursing discipline.According to the knowledge, attitude, belief, practice model and motivation theory, SML can be regarded as a form of endogenous motivation and enhance the faith and values of vocational college nursing students, thus improving their altruistic prosocial behaviour.It is reported that good prosocial behavior can not only stimulate nurses' social responsibility and strengthen their social functions, but also promote the improvement of people-oriented nursing service quality and meet people's increasing demands for diversified and multi-level medical and health services.

3.Why in introduction about students’ characteristics? I have not clear the reasons for these information?

Response: Thank you,which is also something I considered before.I referred to a lot of literature, and other literature reported the demographic characteristics in this way,and they were very detailed.Based on previous literature,prosocial behavior is a dependent variable,it will be affected by demographic factors,in addition to the source of meaning in life and self-efficacy.So,when we think the the mediating value of self-efficacy ,we should consider other demographic factors.

4.You had hypothesized a relation between general self-efficacy, the source of meaning in life (SML) and prosocial behaviours and for these reasons you had tested the model. But on the basis of what? Are there previous studies?

Response: Thank you for your very good suggestion. Based on previous literature,the knowledge, attitude, belief, practice model and motivation theory and rules of the mediating effect,we hypothesized a relation between general self-efficacy, the source of meaning in life (SML) and prosocial behaviours,and our objective is to examine the mediating effect of general self-efficacy on the relationship between the source of meaning in life (SML) and prosocial behaviours in vocational college nursing students.And some literature has explored the mediating effect of life meaning of college students(or adolescents) between family cohesion, nostalgia, the moral sense of life and post-traumatic stress disorder and prosocial behaviors. While few researches explore the mediating effect of self-efficacy, the source of meaning in life (SML) and prosocial behaviors. 

5.You said “Studies show that adolescents with higher prosocial behaviour are more responsible and sympathetic than those without such behaviour, are good at perspective taking, and have relatively high levels of prosocial moral reasoning” but in your study you are speaking about nursing students and not adolescents. Are there other studies about student’s population? about health care workers?

Response: Thank you for your valuable suggestions.At present, there are very few studies on prosocial behavior, most of which are about students or adolescents.Some investigations is for nurses,but the nurses’ values are basically stable,but vocational college students are generally between the ages of 17 and 24 and are mostly new graduates from middle or high school. The majority of them are from only-child families, female, and unmarried. They have relatively unique advantages but also lack positive motivation, have poor communication skills, lack a strong sense of self, have weak teamwork ability, and have poor self-bearing and poor self-management ability .In addition, these students are in an important period in the formation of their life and professional values, so it is very important to cultivate their knowledge and skills, provide them with a quality education and cultivate their humanistic care ability.

6.In page 1, line 31 there are two and.

Response :Thank you for such a detailed suggestion, which is caused by my negligence. We have deleted a and.

7.About the instruments there are only few information. GSES come from WANG but in which kind of population was it validated? In students? Nursing students? If not, there is the necessity to validate the GSES in nursing students. Can we have some information about explorative factor analysis with the identification of the dimensions and about confirmative factor analysis? You used the total score of the GSES but there is a secondary order factor? At the same way it should be reported information about SMLS and PBS?

Response :Thank you ,the GSES come from WANG but it was validated in all kinds of population,such as primary school students, college students, university teachers and hospitalized patients, etc.In different studies, its reliability is slightly different. In this study, its reliability is 0.903.

We rechecked the original version of the WANG,the GSES were single-dimensional,and it’s universal across cultures.And the reliability of SMLS and PBS was 0.978 and 0.962, respectively.

We showed that the results sf explorative factor analysis with the identification of the dimensions and about confirmative factor analysis.We showed the final dimensions of each questionnaire.

8.In table 1 there are the differences in prosocial behaviour and demographic characteristics. Why we have not the same information about general self-efficacy and SLM?

Response :Thank you for your valuable advice, because our objective is to examine the mediating effect of general self-efficacy between the source of meaning in life (SML) and prosocial behaviours in vocational college nursing students and the effect size,so the factors of prosocial behaviors ,including demographic characteristics are very significant.But the demographic characteristics of self-efficacy and SLM have not very significant meaning to the overall results.

9.The paper need more critical discussion. Which is the implication of these finding? Can you work on general self-efficacy to improve the humanistic care quality among nursing students? in which way?

Response :Thank you for your suggestions,we are grateful for the valuable and detailed suggestions. We really don't have a specific description about how to improve the humanistic care quality among nursing students by self-efficacy.Your suggestion gives us a clearer idea.So we added the discussion in page 18,line 331-337.For example,we could improve our teaching methods, such as increasing scene teaching method can improve students' situational coping ability, self-confidence and empathy ability .Meanwhile ,we encourage to increase more humanities courses to increase nursing students' awareness of humanistic care and improve the quality of humanistic care.Moreover, some relevant lectures should be hold regularly to enlighten nursing students' critical thinking and improve their confidence through the power of a good example.

10. In pag 15, line 270 there is ERROR??

Response :Thank you very much for your careful advice to my article. It was our negligence, and we has corrected it now. 

Reviewer #2

1.please check the 270 line what the error means?

Response :Thank you for your comments on this study,It was our negligence, and we has corrected it now. 

2.and the reference list - the correctness of writing and editing

Response :Thank you ,we have do some correctness of writing and editing according to the styles of the PLOS ONE.

We appreciate for your warm work earnestly, and hope that the correction will meet with approval. 

Once again, thank you very much for your suggestions.

---

## [Editor Report · Decision Letter 1]

26 Nov 2020

Mediating effect s  of general self-efficacy on the relationship between the source of meaning in life and prosocial behaviours in vocational college nursing students:a cross-sectional study

PONE-D-20-15550R1

Dear Dr. Chen,

We’re pleased to inform you that your manuscript has been judged scientifically suitable for publication and will be formally accepted for publication once it meets all outstanding technical requirements.

Kind regards,

Giampiera Bulfone

Guest Editor

PLOS ONE

Additional Editor Comments (optional):

Thank for your revision. Every point meet my suggestion.
---

## [Editor Report · Acceptance letter]

4 Dec 2020

PONE-D-20-15550R1 

Mediating effects of general self-efficacy on the relationship between the source of meaning in life and prosocial behaviours in vocational college nursing students:a cross-sectional study 

Dear Dr. Chen:

I'm pleased to inform you that your manuscript has been deemed suitable for publication in PLOS ONE. Congratulations! Your manuscript is now with our production department. 

Kind regards, 

on behalf of

Dr. Giampiera Bulfone 

Guest Editor

PLOS ONE